# Tetrahydrocannabinol and Cannabidiol for Pain Treatment—An Update on the Evidence

**DOI:** 10.3390/biomedicines12020307

**Published:** 2024-01-29

**Authors:** Kawthar Safi, Jan Sobieraj, Michał Błaszkiewicz, Joanna Żyła, Bartłomiej Salata, Tomasz Dzierżanowski

**Affiliations:** Palliative Medicine Clinic, Medical University of Warsaw, Żwirki i Wigury 61, 02-091 Warsaw, Poland

**Keywords:** cannabinoids, nabiximols, tetrahydrocannabinol, cannabidiol, neuropathic pain, nociceptive pain

## Abstract

In light of the current International Association for the Study of Pain (IASP) clinical practice guidelines (CPGs) and the European Society for Medical Oncology (ESMO) guidelines, the topic of cannabinoids in relation to pain remains controversial, with insufficient research presently available. Cannabinoids are an attractive pain management option due to their synergistic effects when administered with opioids, thereby also limiting the extent of respiratory depression. On their own, however, cannabinoids have been shown to have the potential to relieve specific subtypes of chronic pain in adults, although controversies remain. Among these subtypes are neuropathic, musculoskeletal, cancer, and geriatric pain. Another interesting feature is their effectiveness in chemotherapy-induced peripheral neuropathy (CIPN). Analgesic benefits are hypothesized to extend to HIV-associated neuropathic pain, as well as to lower back pain in the elderly. The aim of this article is to provide an up-to-date review of the existing preclinical as well as clinical studies, along with relevant systematic reviews addressing the roles of various types of cannabinoids in neuropathic pain settings. The impact of cannabinoids in chronic cancer pain and in non-cancer conditions, such as multiple sclerosis and headaches, are all discussed, as well as novel techniques of administration and relevant mechanisms of action.

## 1. Introduction

While there continues to be an increase in investigations of herbal cannabis use in relieving HIV and cancer symptoms and ameliorating spasticity disorders, epilepsy, and Tourette’s syndrome, no clear and established approval for their inclusion in medical preparations has yet been accepted. Of particular relevance is cannabinoids’ role in relieving neuropathic pain in cancer settings. Despite growing evidence for this, the International Association for the Study of Pain (IASP) clinical practice guidelines (CPGs) of 2015 provide weak recommendations against such use [1]. These guidelines, however, are based on a mélange of inconclusive data, yet disregard the existing theoretical reasoning and evidence behind cannabinoids’ action and potential benefits. Moreover, this body of evidence and the newly published studies were made publicly available recently, only following the release of the latest IASP CPGs, leaving little consideration for their medical impact and clinical utility. Additionally, it was suggested that cannabinoids, particularly nabiximols (an extract of *Cannabis sativa* containing therapeutic cannabinoids: D9-tetrahydrocannabinol (THC) 27 mg/mL and cannabidiol 25 mg/mL), at high doses, demonstrated effectiveness as analgesics for cancer pain [2]. This postulated benefit, however, was limited by adverse effects of cognitive impairment and dizziness imparted by an elevated dosage. Two other randomized double-blind studies also drew attention to the analgesic effects of nabiximols in advanced cancer patients with pain unsuccessfully managed with opioids [3,4]. Hence, the European Society for Medical Oncology (ESMO), as elucidated in its guidelines, concludes that the effects of nabiximols as an add-on to opioid therapy in advanced cancer pain remains unclear and there is a need for further prospective double-blind placebo-controlled trials with a larger sample size to determine the efficacy and therapeutic dosage. Additionally, what continues to make cannabinoids an attractive option for researchers is the absence of respiratory depression, allowing their combined administration with opioids for synergistic effect, as well as successful pain relief in chemotherapy-induced peripheral neuropathy (CIPN) in animal models [5]. Moreover, systemic reviews and meta-analyses provided evidence that there was no association between all-cause mortality and cannabis use, contrary to opioid use, which resulted in significant mortality, highlighting the relative safety level of cannabinoids [6,7].

As there remains little evidence of cannabinoids acting as potential adjuvants to opioid therapy to relieve cancer and non-cancer pain at low and medium doses [8], this paper aims to review the most recent, relevant, and up-to-date evidence from 2016 leading to 2022, adding to the discussion.

The objectives of this review are as follows:To consider and discuss the mechanisms and types of pain relief induced by cannabinoids;To evaluate the potential of cannabinoids in chronic cancer pain and neuropathic pain with reference to modern existing analgesic modalities;To highlight the most recent updates in preclinical and clinical studies and their impact on researcher consensus.

## 2. Mechanisms of Action in Pain Relief

The cannabinoid receptor, CB1, is found in both peripheral and central nervous systems and is situated on neuronal presynaptic axon terminals, involved in neuromodulation and influencing the behavior of glia, neurons, and microglia [9]. Once activated, these receptors, being G protein-coupled receptors (GPCRs), inhibit adenylyl cyclase, and halt cAMP production [10]. This mechanism triggers the opening of unique G protein-coupled inwardly rectifying potassium channels and the closing of voltage-gated sodium channels, attaining a state of neuronal hyperpolarization, blocking electrical conduction within neurons and inhibiting synaptic transmissions [11]. CB1 receptors are found in highly concentrated amounts in the hippocampal formation and the olfactory bulb, justifying their impacts on memory, cognition, smell, and pain perception [12]. Apart from this, these receptors are found in high amounts in the ventral posterolateral nucleus of the thalamus, the periaqueductal gray matter of the midbrain, the nucleus tractus solitarius, and the dorsal horn of the spinal cord, where GABAergic neurons also reside, contributing to pain perception [13]. CB1 receptors reside in the thick primary afferent Aβ and Aẟ fibers, suggesting their analgesic potential in nerve dissection injuries [14]. Due to limited amounts of CB1 receptors in the brain stem, their impact on the respiratory center is much milder than opioids. CB2 receptors, on the other hand, are found outside the nervous system, with the exception of glia, and are expressed in solid tumor tissues, signifying their potential anti-cancer actions [12,15,16,17] Both CB1 and CB2 receptors are also located in the gastrointestinal tract, taking part in the regulation of its motility and immunological processes due to affecting the microbiome [18]. The endocannabinoid system has at least two endogenous ligands that are currently recognized, namely anandamide (AEA) and 2-arachidonoyl-glycerol (2-AG) [19].

Pain transduction mechanisms rely on the conversion of nociceptive stimuli to neuroelectric signals, which are then transmitted to the somatosensory cortex and brain stem through ascending and descending modulatory pathways [20,21]. While the ascending pathway is responsible for the transmission of the pain signal, the descending pathway acts to modulate and inhibit the ascending pathway. The descending pain pathway consists of a circuit of neurons between the periaqueductal grey matter of the midbrain, the nucleus raphe magnus of the medulla, and the dorsal horn of the spinal cord. The periaqueductal grey matter expresses CB1 receptors, which once stimulated activate the descending pathway to inhibit pain and inflammation. Cannabinoids’ stimulation of the descending pathway is a result of inhibition of presynaptic neurotransmitter and neuropeptide release, hindering neuronal excitability in the ascending pathway and thereby reducing pain signaling [20]. Animal studies have also found a link between the CB1 receptor-mediated stimulation and activation of TRPV1 (transient receptor potential vanilloid type 1) receptor, resulting in “the stimulation and inhibition of OFF and ON cell activity” in transmitting pain stimuli [22]. On a cellular level, the analgesic effect of THC is attributed to complex multimodal mechanisms of action via cannabinoid receptors CB1 and CB2 and its interference with a series of other pathways, among which are delta and kappa opioid receptors, the GABA-ergic/glutamatergic system, and the noradrenergic system [10]. When the CB1 receptor on the presynaptic terminal is activated, it leads to the inhibition of voltage-gated calcium channels and the cAMP/PKA pathway. These processes result in a reduction in the release of neurotransmitters, which, in turn, lowers pain transmission and perception [23]. Additionally, CB1 receptors are also responsible for synaptic plasticity and can form homo- or hetero-dimers with various other classes of G protein-coupled receptors, including opioid or alpha-2 adrenergic receptors [24]. Through close interaction of the cannabinoid and opioid signaling pathways, a synergistic mechanism of action is exhibited with the analgesic effect of THC, aiding in altering pain perception [25]. Moreover, antinociceptive properties are enhanced by stimulating norepinephrine release from descending inhibiting pathways [26]. By acting as transient receptor potential vanilloid 1 (TRPV1) agonists, cannabinoids also play a role in blocking hyperalgesia, allodynia, and thermal and mechanical stimuli [26]. The binding of cannabinoids to their receptors impacts signaling pathways including those related to calcitonin gene-related peptide (CGRP), responsible for vasodilation, nuclear factor kappa-light-chain-enhancer of activated B-cells (NF-κB), which regulates host immune response in the face of infection, G protein-coupled receptor 18 (GPR18), inducing apoptosis of proinflammatory macrophages [26] and peroxisome proliferator-activated receptors (PPARs), nuclear transcriptions factors that also play a role in modulation of pain sensation [27].

Activating PPARs α and γ or the TRPV1 ion channel is responsible not only for the anti-inflammatory and analgesic effects but also for their anti-tumorous and cardiovascular-protective functions [28]. The effect of cannabinoids on adenosine, serotonin, and dopamine receptors has also been demonstrated [27]. Although recent evidence suggests that cannabinoids binding in the midbrain cause the release of dopamine from the neurons of the central tegmental area, resulting in relief and pain reduction, such a finding remains controversial, as CB2 receptor expression in the midbrain has not been confirmed [27]. Through non-receptor actions, however, cannabinoids are also capable of inhibiting cyclooxygenase (more COX-2 than COX-1), and blocking the production of inflammatory and pain mediators at the location of tumors, which explains their anti-inflammatory action [26].

## 3. Medical Use and Metabolism

Non-prescribed cannabis is most commonly inhaled through vaporization at a temperature of 175–225 °C. Oral administration in the form of tablets is the most common route used for medical purposes, although it has a fourfold decrease in bioavailability in this form [29,30]. Metabolism occurs predominantly through hepatic first-pass metabolism by cytochromes P450 2C9, 2C19, and 3A4, which convert THC to the most metabolic and psychoactive form, 11-hydroxy-Δ9-THC, and the inactive byproduct 11-nor-9-carboxy-Δ9-THC [31]. The bioavailabilities of the oral forms are around 10% and 30–40% for inhaled preparations [30,32]. More accurately, via the inhalation route, the peak plasma concentration of THC is reached in several minutes, with a Cmax 70 ng/mL, whereas in oral supply, the peak plasma concentration of THC is reached at 2–6 h after ingestion with a Cmax 1.8 ng/mL [30]. Additionally certain routes of administration relieve certain types of pain. THC and CBD via the oromucosal route, for example, are suitable for neuropathic and cancer pains, whereas THC in the oral form provides promise for nociceptive pain. THC inhalations are also effective in neuropathic pain [33]. 

THC is highly lipophilic, hence effectively crossing the blood–brain barrier, accumulating in tissues with a high volume of distribution and a slow elimination mechanism [34]. After inhalation, it takes 15 min to reach maximal brain concentrations and present somatic and psychic symptoms [34]. This state is sustained over the following 2–4 h and gradually disappears [34]. With oral administration, peak levels are achieved after 30–120 min, at which time effects appear, lasting from 5 to 12 h. The concentrations in the brain are much higher after inhalation [35]. The addition of CBD (cannabidiol) to THC has been shown to enhance the positive effect of THC and reduce its side effects [4]. Furthermore, CBD affects the metabolism of THC, which prolongs its effect [36,37,38]. This dependence may explain the superiority of oromucosal nabiximols spray to synthetic forms of THC such as dronabinol [4].

## 4. New Preclinical Evidence

### 4.1. Pain, Migraine, and Headaches

A recent systematic review considers the role of cannabinoids in the treatment of pain, migraines, and headaches [39]. The study summarizes data from a meta-analysis of 17 preclinical studies. It provides good-quality evidence for the synergistic action of opioid and cannabinoid co-administration in relieving the nociceptive pain of headaches and migraines, which resulted in a 3.6-fold decrease in the median effective dose (ED50) of morphine administered with THC, compared to the ED50 of morphine alone [40]. A similar effect was witnessed with codeine, wherein the ED50 of codeine with THC was 9.5 times less than that of codeine [40,41]. Not only does this decrease in the required dosage increase potency and decrease the risk of adverse events, but there is thorough evidence of an opioid-sparing effect with cannabinoid use. In animal studies with chronic neuropathic pain, the effect of morphine lasted up to 22 days, after which drug tolerance occurred. This effect was not observed in rats treated with THC and CBD [36,42]. Another intriguing phenomenon observed in animal studies is that cannabinoids’ anti-inflammatory properties were several hundredfold more potent than aspirin [43]. It has been demonstrated that people suffering from headaches may also have impaired functioning of the endocannabinoid system. This suggests that cannabinoids could be an effective drug in this indication. An additional advantage is their potential effect on symptoms that often accompany migraines, such as nausea, vomiting, and anxiety [44].

### 4.2. Effects on Nausea, Vomiting, and Appetite

Chemotherapy-induced nausea and vomiting (CINV) is a common side effect present in cancer patients during therapy. Despite using high doses of antiemetics, CINV is still troublesome for many patients [45]. It has been supported in animal models that cannabinoids induce antiemetic effects [45]. It is suggested that cannabinoid-mediated acetylcholine blocking causes digestive tract motility inhibition [46]. Another suggestion is that cannabinoids bind to CB1 receptors located in the dorsal–vagal complex of the brain stem, the critical region where the emetic nuclei are located and where afferent stimuli from the upper gastrointestinal tract arrive, evoking nausea and emesis. Both dronabinol and nabilone have been approved to treat chemotherapy-associated nausea and emesis [4].

Major weight loss is another frequently witnessed issue in advanced cancer patients. Experimental studies exposing animals to delta(9)-tetrahydrocannabinol confirm that THC and other cannabinoids have stimulatory effects on appetite and can boost food consumption [46,47]. Moreover, dronabinol has been approved for AIDS wasting syndrome. Benefits have also been documented in terminally ill patients, particularly those prone to depression and anxiety [46,47].

### 4.3. Significance of Preclinical Studies

A review of the preclinical studies provided a comprehensive overview of cannabinoids, cannabimovones (CBMs), and endocannabinoid system modulators in terms of their antinociceptive efficacy in animal models of persistent and injury-related pain [25]. The up-to-date preclinical evidence substantially supports the hypothesis of analgesia through cannabinoid administration. A recent brain imaging study (from 2020) demonstrated that inhaled vaporized cannabis containing THC (10.3% THC; 0.05% CBD) aided in uncoupling the “resting-state functional connectivity” in the raphe nuclei of rats treated with paclitaxel [25,48]. This helped normalize the hyperconnectivity induced by paclitaxel and yielded antinociceptive effects [25,48,49]. Analysis of the preclinical studies demonstrates value in explaining the drug mechanisms, and identifies drug impacts at well-defined doses in monitored environments with rather predictable pharmacokinetics [25]. For example, rodent models have been used to test the effects of synthetic cannabinoids (SCs) on inflammation and syndromes related to the cardiovascular and respiratory systems, cancer treatments, and metabolic conditions.

In terms of analgesic effects, behavioral experiments have shown significant preclinical evidence of SCs’ effectiveness in various pain models [49,50,51,52]. However, the majority of research has been centered around chronic pain states, wherein SCs have demonstrated effectiveness once again, particularly in models of neuropathic and chronic inflammation [53,54,55,56,57,58,59,60,61,62]. One possible mechanism of action involves the inhibition of mast cell degranulation and neutrophil migration by activating CB2 receptors, which can result in a decrease in inflammation [59,63,64]. Adverse effects of SCs overdose include cardiac toxicity, gastrointestinal changes, acute rhabdomyolysis, malignant hyperthermia, stroke, and seizures [65,66,67]. Long-term effects include an increased risk of myocardial infarction and cognitive impairment [68].

These animal model studies also focus on a diverse variety of cannabinoids and endocannabinoid systems, many of which have never been tested on humans [25]. Collectively, more than 150 preclinical animal studies addressing CB2 agonists, reuptake inhibitors, monoacylglycerol lipase (MGL), and fatty acid amide hydrolase (FAAH) inhibitors in pain relief exist, but clinical randomized controlled trials assessing their efficacy are largely non-existent. Despite promising results from animal studies, the effects on the ECS are still unpredictable. A clinical trial involving a drug called BIA10-2474, an inhibitor of FAAH (an enzyme that breaks down endocannabinoids) led to serious side effects, including the death of one person [36,69]. This is significant, as BIA10-2474 failed phase I clinical trials in 2016 due to its serious adverse effects, which are attributed to numerous off-target actions. FAAH enzymes are classified as serine hydrolases, whose actions in the body are not fully understood [70]. It is also speculated that drug engagements at other protein sites occur, yielding unpredictable adverse effects. Clinical trials with other FAAH inhibitors, however, did not result in serious side effects [71].

## 5. Recent Randomized Controlled Trials and Other Observational Studies

### 5.1. Cannabinoid Use in Cancer Patients

A recent rigorous systematic review and meta-analysis of six randomized controlled studies performed according to the Preferred Reporting Items for Systemic Review and Meta-Analysis Protocol (PRISMA) examined the effect of cannabinoids on cancer pain [72]. Four of these studies assessed THC, while the remaining studied nitrogen-containing benzopyran derivatives, such as delta-1-trans-tetra hydrocannabinol (NIB). All phase III trials discussed in the review revealed no true analgesic benefit from cannabinoids, while one phase II study witnessed a decrease in pain scores with cannabinoids [3]. The effect on the absolute change in mean pain intensity was assessed using the Numeric Rating Scale (NRS) pain scores and compared the outcomes to a placebo [72,73]. The results suggested that cannabinoids are not effective in reducing cancer pain, with the cannabinoid treatment group experiencing significant adverse effects and higher participant dropouts. Cannabinoids were also assessed as clinically relevant adjuvants to opioid medications in advanced cancer pain of different etiologies [8]. The findings provide encouraging evidence for low- and medium-dose cannabinoids in this setting, with no serious adverse effect at such dose ranges. The author agrees, however, that there is insufficient high-quality evidence to derive a more firm conclusion [8]. The WHO guidelines for management of cancer pain in adults and adolescents suggest that more data analysis is needed on the impact of cannabinoids in these patients [74]. Not only that, but the present review highlighted that cancer patients do use cannabinoids. Through a series of anonymous surveys, for example, it was seen that 18% of cancer patients in Canada used cannabinoids within 6 months of the survey [72]. Forty-six percent of those justified their use by referring to cancer-related pain. In another study, where urine analysis from the American states was examined, 21% of those tested were positive for cannabis use within the last month, most commonly for cancer pain [75]. 

A double-blind randomized clinical trial was conducted to measure the impact of nabiximols (a novel cannabis extract) on pain in advanced cancer patients presenting with opioid-resistant pain [46]. The results indicated a successful improvement in the patients’ pain sensation when the cannabis extract was administered, yet again highlighting the role of cannabinoids as promising analgesics for cancer patients. The systematic review and meta-analysis by Boland et al. aimed to determine their beneficial effects in the treatment of cancer-related pain [72]. An oromucosal spray containing 1:1 THC:CBD extract (nabiximols) was used in those studies. Dose titration varied between studies, with some containing a fixed dose and others requiring patients to self-titrate for optimal management. Phase III studies set out a specified dose escalation protocol for pain relief achievement, response to adverse events, or until a maximum dose of 10 sprays per day was reached. The study concluded that cannabinoids added to opioids did not reduce cancer pain in adults. The meta-analysis revealed statistically significant higher odds of dizziness and somnolence in the cannabinoid group. A strength of this study is its low risk of bias.

Another cross-sectional survey of 926 patients suggested that 75% of cancer patients seek information about cannabis use in response to their disease, yet only 15% receive it from their cancer teams [75]. Furthermore, only 30% of oncologists in the United States claim to be appropriately trained to give professional cannabis recommendations [76]. The first placebo-controlled, double-blind, randomized clinical trial to assess the efficacy and safety of CBD in advanced cancer patients is being led by Good et al., with results unavailable so far [77]. Based on the up-to-date data and presented studies, it can be concluded that the overall stance of cannabinoids use in cancer pain settings is rather weak, with cannabinoids not appearing to have any real impact as adjuncts to opioids or when opioids have failed to relieve pain. Given that most studies are of moderate or weak quality and with rather inconclusive or weak conclusions, more research will be needed to confirm the existing assumptions.

### 5.2. Randomized Controlled Trials with Non-Cancer Patients

A double-blind, placebo-controlled trial has demonstrated the effectiveness of nabilone in relieving the symptoms of medication overuse headaches [78]. Additionally, it was found to be more effective than ibuprofen. In a survey study of 139 chronic cluster headache patients by Leroux et al., 45.3% of patients reported using cannabis to manage their pain, and 25.9% confirmed a positive effect [44,79]. However, there are not enough double-blinded, placebo-controlled trials to determine the true effect of cannabinoids on various types of headaches [44].

The Annals of Plastic Surgery published a systematic review of the PubMed database, focusing on the role of cannabinoids in pain. Twelve primary studies were discussed, including prospective cohort studies as well as randomized controlled trials (RCTs) on both animals and humans. The findings concluded that oral and topical CBD have shown “early positive” results with negligible side effects; however, the evidence is based on too small of a sample size to generalize the findings and deduce a larger, more meaningful impact [80]. A double-blinded RCT by Nitecka-Butcha et al. studied the impact of topical CBD on myofascial pain [81]. A 65 mg topical dose of whole hemp extract was applied to the masseter muscle region twice daily for 2 weeks in the experimental group. The results revealed a 70% reduction in VAS scores, from the average VAS score of 5.6 to 1.7, versus a 9% decrease in the placebo group [81]. Another RCT examined topical CBD action in the setting of lower extremity neuropathy [80]. An amount of 250 mg of pure CBD was used for 4 weeks, four times daily, and later patients switched to a placebo treatment for 4 weeks. The Neuropathic Pain Scale scores for intense and sharp pain were reduced significantly in the treatment group (*p* < 0.001).

### 5.3. A Novel Selective-Dose Cannabis Inhaler in Patients with Chronic Pain

Even though smoking cannabis triggers a rapid onset of effects, the Δ9-THC plasma concentrations remain volatile, which indicates inconsistent delivery of active compounds. Likewise, smoking cannot be approved for therapeutic purposes, due to the health hazards evoked by pyrolytic byproducts [31]. The development of a certified, accurately temperature-controlled device for vaporizing dry cannabis flowers is essential to safely deliver active compounds to the patients [82]. In a randomized, three-armed, double-blinded, placebo-controlled, cross-over trial, 27 patients were administered with a single inhalation of Δ9-THC: 0.5 mg, 1 mg, or placebo [83]. The Δ9-THC plasma levels were examined prior to the inhalation, as well as up to 150 min after the vaporization. The pain sensation was assessed on a 10 cm visual analogue scale. The Syqe Inhaler (PitchBook, Seattle, WA, USA), a portable, battery-powered, software-controlled (Novachem, Victoria, Australia) medical device, was used during the study. The two-second heating process is activated by inhaling. By that time, 90% of the inactive acidic THC undergoes decarboxylation to become an active THC form. The aerosol is then inhaled by the patient. The largest decrease from baseline in the VAS pain score was observed in the patients that were administered with a 1 mg dose of Δ9-THC, while the placebo group had the lowest decrease. The difference between both groups was statistically significant. In all study groups, the maximum number of patients experiencing a reduction of 30% or more in the pain VAS score was reached 120 min after inhalation. Adverse effects were mild and self-resolving. No indication of negative impacts on cognitive performance was noted. The development of this medical vaporizing device allowed the delivery of active compounds selectively and precisely with controlled side effects.

### 5.4. The Prevalence of Medical Cannabis Use by Patients with Chronic Pain

An American cross-sectional study, conducted from 2011 to 2015, identified around 10.3 million chronic pain patients, out of which 247,949 individuals reported cannabis use. The number of patients using cannabis increased from 33,189 in 2011 to 72,114 in 2015. Moreover, the yearly proportion of chronic pain patients using cannabis increased from 2.4% in 2011 to 3.9% in 2015 [84]. The average age of cannabis users increased from 42.80 to 45.40 years, with the highest proportion of users being aged between 45 and 64 years old. Female patients had a lower proportion of cannabis use. Patients in the lowest income quartile had the highest proportion of cannabis use (39.4%), while those in the highest quartile had the lowest proportion (11.9%) [84]. The authors noted an upward trend in the prevalence of cannabis use among patients with chronic regional pain syndrome (1.25% to 1.96%), trauma (5.23% to 8.98%), spondylosis (0.57% to 1.2%), failed back surgery syndrome (0.47% to 0.97%), and other chronic pain conditions (2.4% to 3.49%) [84]. 

The overall hospital expenses for patients who used cannabis increased from USD 31,271 (with a standard deviation of USD 1333) in 2011 to USD 38,684 (with a standard deviation of USD 946) in 2015 [84]. Baron et al. discovered that in all states where medical cannabis was legalized, pain syndromes including migraines, headaches, and arthritis were the most frequently reported reasons for cannabis use [39]. On a global scale, medical cannabinoid use is legal in 40 countries and 38 states of America [85]. It was also found that cannabis was most commonly used as a replacement for opioid medications, followed by anti-depressants, anti-anxiety drugs, and nonsteroidal anti-inflammatory drugs (NSAIDs) [39]. Andreae et al. [86]. reviewed six randomized controlled trials of phytocannabinoids and calculated a number needed to treat (NNT) of 5.6 for 30% neuropathic pain reduction, whereas Petzke et al. [87]. found an NNT of 14.

## 6. Recent Systematic Reviews

Wang et al. (2021), in a systematic review with a meta-analysis of 32 randomized controlled trials, including 5174 patients, discussed the use of medical cannabis and cannabinoids in chronic non-cancer- and cancer-related pain [88]. The a priori subgroup hypothesis addressed the treatment effects in association with the following:Chronic non-cancer pain versus chronic cancer pain;Neuropathic pain versus non-neuropathic pain;THC alone versus THC and cannabidiol (CBD) versus CBD alone versus palmitoylethanolamide (PEA);Inhaled versus ingested versus topical cannabis;Enriched enrolment versus non-enriched;High versus low risk of bias;Industry-funded versus non-industry-funded trials.

The pool of patients with chronic non-cancer pain entailed neuropathic pain, spasticity-related pain, nociplastic pain, nociceptive pain, medication overuse headache, and mixed chronic non-cancer pain. The average baseline pain score on the 10 cm VAS scale was 6.5. Moderate-quality evidence demonstrated that non-inhaled cannabis resulted in a small increase in pain relief above or at a minimally important difference (MID) value of 1 cm on the VAS scale. A higher number of patients also experienced >30% pain reduction with medical cannabis compared to the placebo (relative risk (RR):1.21, 95% CI 1.004 to 1.47; RD 7%, 0.1% to 16%). High-quality evidence also revealed that oral medical cannabis resulted in a very small increase in the number of patients improving their physical functioning and attaining an MID of at least 10 points and a weighted mean difference of 1.67 (as measured by the SF-36 physical functioning scale).

To assess the impact on sleep quality, 16 RCTs (randomized controlled trials) were conducted, and their metanalysis concluded that oral medical cannabis resulted in a significant sleep quality improvement [88]. Additionally, emotional functioning was not improved with medical cannabis compared to placebo [4,88,89,90,91,92,93]. The main adverse events were cognitive impairment, vomiting, drowsiness, dizziness, impaired attention, and nausea. Five RCTs in the same systematic review analysis shed some light on cannabinoid-related adverse events. Moderate-quality evidence suggests that transient cognitive impairment is the major adverse effect experienced by patients, with a risk ratio of about 2.39. Meta-regression of the same study revealed a time-dependent increase (from the beginning of cannabinoid use to just under 3 months) in the risk of dizziness with oral cannabinoid administration [88].

When comparing the analgesic abilities of non-inhaled medical cannabis and cannabinoids to the standard, modernly used pain relievers, the group of non-steroidal anti-inflammatory drugs (NSAIDs), the collected results are rather conflicting. One trial concluded that PEA was less effective than celecoxib for nociplastic and pelvic pain relief in women, while a crossover trial examining the effectiveness of THC and ibuprofen for headaches suggested no significant difference [88,94]. Moreover, no significant difference was obtained from the data related to physical functioning, emotional and mental functioning, vomiting, dizziness, impaired attention, and nausea [88]. Low-certainty evidence collected highlighted that in chronic neuropathic pain, there was no difference in analgesic potential between nabilone and dihydrocodeine [95]. This systematic review provided moderate- to high-quality evidence demonstrating that non-inhaled medical cannabis or cannabinoids, compared with placebo, resulted in a small to very small proportion of increase in pain relief, sleep quality, and physical functioning, alongside several adverse effects, in both chronic cancer and non-cancer pain. Overall, no difference in the analgesic effect for neuropathic versus non-neuropathic pain for chronic cancer and non-cancer pain was found in the treatments [88].

Medical cannabinoids have not only been found to impart a subjective improvement in pain perception, but also provide symptomatic relief, particularly in relation to sleep disturbance, appetite disorders, and nausea [96]. According to patient reports, they also improve the ability to focus and overall functioning. An important aspect of the use of cannabinoids in chronic pain is the patient’s attitude toward the therapy. Zeng et al. address this issue in a systematic review, whereby factors influencing patients’ decisions are considered. The positive effect and the reduction in opioid intake make patients more willing to accept treatment with cannabinoids. However, the fear of addiction, lack of control, and the potential negative attitude of family and loved ones have the opposite effect. Other important factors include the age and legal status of the patient, cost, and availability of therapy. Most patients, especially those with advanced life-limiting diseases, preferred oral over inhaled formulations. By contrast, patients who supported the use of cannabinoids for both medical and recreational purposes were more likely to smoke cannabis. The diversity in the composition of the preparation used was also interesting. People who supported only the medical use of cannabis preferred preparations with a high concentration of CBD or similar CBD:THC ratios. In contrast, patients who also admitted to recreational use were more likely to use products with a high THC content, which is associated with more side effects.

A systematic review addressing cannabinoid use for lower back pain in the geriatric population analyzed 23 articles and described the benefits and limitations of CBD and THC, highlighting the lack of sufficient data in the field [97]. Lower back pain remains a significant complaint, for which cannabinoids have been reported to be the most-used substance, in adults and the elderly, with an American cross-sectional study revealing that 46% of individuals who previously used cannabis recreationally admitted to using it for pain management [98]. The authors conclude that further research is necessary to consider cannabinoids in patients’ management plans [97]. Additionally, a multicenter study conducted over 38 weeks involved a group of 380 patients using a mucosal THC/CBD mixture (in spray form), for peripheral neuropathic pain. The results revealed a clinical improvement in at least 30% of the experimental group when compared to the placebo [99]. Patient-centered data collected by Bruce et al. corroborates these findings, highlighting that medical cannabis use consistently revolved around three main reasons: 1. as an alternative option to over-the-counter options; 2. complementary to prescribed medicines; 3. in assisting with tapering off a prescribed drug [100].

Another unique setting in which cannabinoids may seem to have potential is HIV-associated neuropathic pain. A systematic review by Aly et al. sheds light on the impact of cannabidivarin (CBDV) and discusses the existing limited evidence from three RCTs and 10 preclinical studies. The findings highlighted this drug’s safety but noted its poor action due to a lack of receptor affinity [101]. However, although not a cannabinoid, an alternative FDA-approved terpene and CB2R-selective agonist, namely β-caryophyllene, is more specific and offers a more reliable mechanism of action, as well as having demonstrated effectiveness in preclinical studies, and is, therefore, hypothesized to be of therapeutic benefit in HIV neuropathy and worthy of future evaluation and study [102,103]. Other agents deemed to be effective in HIV neuropathy, as per the authors’ perspectives and clinical data, are FAAH (fatty acid amide hydrolase) inhibitors and IPM (indomethacin plus minocycline) regimens [101]. A summary of the most significant findings with an overview of the relevant guidelines and future directions in the research on cannabinoids for pain are provided in Table 1.

## 7. Current Availability and Approval in Europe

As considerable public and political interest in medical cannabis use has been heightened recently, the task force of the European Pain Federation (EPF) conducted a survey studying the approval status of all cannabis-based medical products and their availability for chronic pain, symptom control, and palliative care. It was found that 21 national EPF chapter representatives approved THC/CBD oromucosal spray for refractory spasticity in multiple sclerosis [110]. Four EPF chapters approved of synthetic THC analogues (nabilone) for chemotherapy-induced nausea and vomiting. Six chapters have an expanded-access program, while German pain societies established medical cannabis as the third-line agent for chronic pain, using the multimodal analgesia approach [110]. However, experts from Finland and the Drug Commission of the German Medical Association advise against the prescription of medical cannabis, attributing their reasoning to insufficient high-quality evidence of efficacy and safety profiles [110].

## 8. Conclusions and Summary

This update review paper aimed to examine the emerging body of evidence addressing cannabinoids as a potential pain treatment. Through their multimodal mechanisms of action, many advantageous attributes have been noted in preclinical studies, case reports, and randomized controlled trials. Although much remains to be investigated, and more clinical trials are needed to outline guidelines for prescription and safe use, this review highlighted possible roles of cannabinoids in neuropathic pain, neuroplastic pain, migraine, and chronic pelvic pain, and it suggested their impact on sleep quality and physical functioning. Furthermore, no significant difference was found between non-cancer and cancer pain relief. The given review also summarizes common and novel routes of administration of cannabinoids and highlights some of the most prominent studies and data present at the levels of preclinical studies, animal studies, and randomized controlled trials. A short account of the prevalence and popularity of cannabinoids is also discussed. To conclude, further studies will be crucial in determining the efficacy and safety of medical cannabis for chronic pain and palliative care symptoms. With the majority of studies revealing marginal benefits or non-significant effects in chronic pain, this aspect of cannabinoid use remains to be evaluated more thoroughly. Larger randomized clinical trials with more patients and bigger sample sizes and research conducted for more protracted time periods (more than 12 weeks) are necessary to help establish current standards and for recommendations to be included in the guidelines and potentially the registry of the European Medical Agency [87,110]. These clinical trials will facilitate the creation of justifiable prescription and safe use guidelines for cannabinoids in the future. Clear overall conclusions remain difficult to arrive at. Experts from the American National Academy of Sciences are of the opinion that insufficient evidence exists regarding cannabinoids’ effectiveness in cancer pain treatment. The studies discussed in this review agree that no significant difference was found between non-cancer and cancer pain relief. Members of the American National Academy of Sciences also agree that cannabinoids are effective for chronic pain in adults, nausea and vomiting resulting from cancer chemotherapy, and spasticity in multiple sclerosis [104]. With continuous evidence being collected, arguments arising to increase the medical use of cannabinoids remain in need of more concrete support.

## Figures and Tables

**Table 1 biomedicines-12-00307-t001:** Relevant guidelines and future directions in the research on cannabinoids for pain.

Year	Relevant Statements and Findings
2015	IASP guidelines announce weak recommendations against cannabinoid use for neuropathic pain in cancer patients [1].
2017	Expert discussion in the American National Academy of Sciences concluded that insufficient evidence exists regarding cannabinoids’ effectiveness in cancer pain treatment, although they are effective for chronic pain in adults, nausea and vomiting resulting from cancer chemotherapy, and spasticity symptoms in multiple sclerosis [104]
2018	ESMO guidelines conclude that there is an unclear role of nabiximols as an add-on therapy in advanced cancer pain [5].
2019	The first placebo-controlled, double-blind, randomized clinical trial on the efficacy and safety of CBD in advanced cancer patients has been initiated, with results unavailable so far [77].
2020	Johal et al. provided a meta-analysis of randomized clinical trials evaluating cannabinoids in chronic non-cancer pain, suggesting that the known evidence is of moderate quality and confidence in this treatment remains low [105].
2021	A pooled meta-analysis of randomized controlled trials on the effect of nabiximols in reducing chronic neuropathic pain concluded that they were superior to the placebo [106].
2023	Clinical trials on cannabinoids in pain-related complaints are emerging across different medical disciplines. Research in the settings of post-ureteroscopy pain, irritable bowel syndrome, and delayed-onset muscle soreness are ongoing [107,108,109].

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
