# Peer review of "Tetrahydrocannabinol and Cannabidiol for Pain Treatment—An Update on the Evidence"

_biomedicines, 2024, doi:10.3390/biomedicines12020307_

Round 1

Reviewer 1 Report

Comments and Suggestions for Authors

The review is generally well written 

There are some typos that need to be fixed within the manuscript 

Several preclinical studies are missing in the references (i.e. those from Di Marzo, Piomelli, Maione). Some of them should be quoted because important in the field of pain 

Comments on the Quality of English Language

Minor revision

Author Response

Thank you very much for taking the time to review this manuscript. Please find the detailed responses below and the corresponding revisions/corrections highlighted in the re-submitted manuscript.  

Reviewer 2 Report

Comments and Suggestions for Authors

Page 1, Lines 9-10: Clarification is needed as to whether cannabinoids are only an attractive pain management option when added to opioid treatment – are they also effective in treating pain on their own?

A brief overview of pain transduction mechanisms in Section 2 would benefit readers to connect how pain is felt, the role of cannabinoids in pain transduction, and then how cannabinoids act to relieve pain.

The underlying premise of the review that the utility of cannabinoids for treating chronic pain is not necessarily accurate.  At a minimum, this assertion is controversial. 

The authors state the expression of CB1R and CB2R are widespread in central and peripheral nervous systems and are found on cell bodies, axons, and dendrites.  This statement has several inaccuracies. First, there is very little evidence of widespread CB2R expression in the nervous system under baseline non-injury conditions.  Second, CB1R, which is found on neurons, is expressed primarily at axon terminals as opposed to dendrites, axons, and/or cell bodies.

The authors state that electrical impulses are conducted from cell to cell but this is not an accurate description of how neurons work.  Communication within neurons is electrical.  Communication between neurons is chemical and occurs via synaptic transmission.

The potassium channels opened by CB1R are a specific type of potassium channel called G protein-coupled inwardly rectifying potassium channels.

Throughout Section 2 on mechanisms of action in pain relief the authors seem to be suggesting that THC and cannabinoids act through several other receptor targets such as CGRP, NF-kB, adenosine, dopamine, and serotonin receptors.  This is not accurate.  While it is true that cannabinoids may modulate these signaling pathways, this occurs via action of cannabinoids at cannabinoid receptors such as CB1R. 

Due to the lack of expression of CB2R in the brain, studies suggesting cannabinoid action on dopaminergic signaling via receptors located in the midbrain are controversial.

In section 3, are there specific routes of administration that are better for treating different types of pain?

The loss of response with repeated cannabinoid administration is typically referred to as tolerance rather than resistance.  It's not accurate to suggest that tolerance for cannabinoids does not occur.

The authors frequently use the term "proven" but seldom does this apply or is it accurate in science. It would be better to say that something was demonstrated, supported, or strongly supported.

In section 4.3, the authors discuss the ability of synthetic cannabinoids to modulate acute pain in preclinical model but both of the papers cited are reviews and did not assess acute pain.  There are other multiple places throughout the review where the authors cite reviews when it would be better to cite the original research showing the findings that they are claiming.

Later in Section 4.3 on the Significance of preclinical studies, the authors discuss toxicity of synthetic cannabinoids in human clinical populations.  The organization of this section is unclear. 

The authors discuss a clinical trial involving BIA10-2474 that resulted in the death of one subject. However, the limited discussion by the authors don't mention important information that this compound was later shown to have MANY off-target effects.  Thus, the lethality observed is not likely due to on-target effects at FAAH.  Especially, given the lack of other serious side effects associated with other FAAH inhibitors that have went through clinical trials. The comment is misleading in a way that could unfairly damage future research and human trials using FAAH inhibitors.

In Section 5.1, the authors discuss a number of different clinical studies [references 63,64,65, 8] without being clear about what the results were from those studies.

At the end of Section 5.1, what is the conclusion regarding where the current literature stands on cannabinoid use in cancer patients.

In Section 5.2, the authors mention RCT being done in animals.  What does this mean?

The statement that medical marijuana in the United States is legal in 29 states is out of date and the number is closer to 38.

There is strange spacing throughout the manuscript.

It is not accurate that beta-caryophyllene is a phytocannabinoid.  It is a terpene that is found in many non-Cannabis plants as well.

The timeline shown in Table 1 seems somewhat random and arbitrary in terms of what is being included.  Many of the items listed were not the most major findings in the field during this time frame.  In particular, the National Academy of Science in the US convened an expert panel to assess medical uses of cannabinoids.  This is not included in the table or anywhere in the review.

A Table of important randomized clinical trials could be provided. 

The purpose of a good review is to provide clarity on topic. This review doesn't necessarily accomplish that goal. The review summarizes recent pre-clinical and clinical studies without providing many clear overall conclusions about whether cannabinoids are effective for pain.  The only real conclusion made is that the authors state more research needs to be done in this area without providing detail in terms of what additional experiments need to be done.  

Comments on the Quality of English Language

The review requires editing for English grammar and typing edits.

Author Response

Dear Reviewer, 

I would like to take the opportunity to thank you for your time and effort in reading and reviewing my manuscript.  Please find the detailed responses below and the corresponding revisions in the re-submitted manuscript.  
